# Generate to Ground: Multimodal Text Conditioning Boosts Phrase Grounding in Medical Vision-Language Models

**Felix Nützel**[1]                                 FELIX.NUETZEL@FAU.DE
**Mischa Dombrowski**[1]                    MISCHA.DOMBROWSKI@FAU.DE
**Bernhard Kainz**[1,2]                       BERNHARD.KAINZ@FAU.DE
[1] *Friedrich-Alexander-Universität Erlangen-Nürnberg, DE*
[2] *Imperial College London, UK*

**Editors:** Accepted for publication at MIDL 2025

## Abstract

Phrase grounding, *i.e.*, mapping natural language phrases to specific image regions, holds significant potential for disease localization in medical imaging through clinical reports. While current state-of-the-art methods rely on discriminative, self-supervised contrastive models, we demonstrate that generative text-to-image diffusion models, leveraging cross-attention maps, can achieve superior zero-shot phrase grounding performance. Contrary to prior assumptions, we show that fine-tuning diffusion models with a frozen, domain-specific language model, such as CXR-BERT, substantially outperforms domain-agnostic counterparts. This setup achieves remarkable improvements, with mIoU scores doubling those of current discriminative methods. These findings highlight the underexplored potential of generative models for phrase grounding tasks. To further enhance performance, we introduce Bimodal Bias Merging (BBM), a novel post-processing technique that aligns text and image biases to identify regions of high certainty. BBM refines cross-attention maps, achieving even greater localization accuracy. Our results establish generative approaches as a more effective paradigm for phrase grounding in the medical imaging domain, paving the way for more robust and interpretable applications in clinical practice. The source code and model weights are available at https://github.com/Felix-012/generate_to_ground.

**Keywords:** Phrase Grounding, Visual Grounding, Stable Diffusion, Latent Diffusion Models, Cross-Attention, Chest X-Rays

## 1. Introduction

Phrase grounding refers to the ability of a model to map textual tokens to regions in an image. Unlike typical object detection or segmentation tasks, phrase grounding usually takes natural language as input, such as medical reports, instead of relying on a predefined set of categories. Thus, phrase grounding can be seen as a generalization of object detection.

In the medical domain, phrase grounding can be used to localize anomalies in images based on textual descriptions provided by experts (Bhalodia et al., 2021). This is attractive, since it works without explicit labels, which are rare and expensive for medical data. By inspecting phrase grounding performance, it is possible to infer which phrases or regions influenced the decision of the system, assess whether the model made proper use of all available modalities, and if the modalities were aligned properly without confusion (Parcalabescu and Frank, 2020). These properties are vital for interpretability, which is a necessary

requirement for models to be used in the medical field (Chen et al., 2023). In addition, without interpretability, models could introduce harmful biases without explanations, which is especially critical for high-risk decision-making (Hakkoum et al., 2022).

Existing discriminative medical phrase grounding approaches can be roughly categorized into two groups: supervised and self-supervised with contrastive learning (Boecking et al., 2022; Gupta et al., 2020; Zhang et al., 2022). An important supervised approach is MedRPG (Chen et al., 2023), which uses ground-truth bounding boxes of radiographs to formulate a contrastive loss based on the features of bounding boxes, as well as the joint attention of bounding boxes, the class token and an additional learnable token. Another relevant branch of medical phrase grounding methods are those that work with 3D medical data, such as the paper by Ichinose et al. (2023), which addresses the unique issues of phrase grounding in CT scans. They suggest using a pre-trained segmentation model that labels the anatomic structures visible in the scan and introduce a module to structure the corresponding medical reports. However, such supervised methods require ground-truth bounding boxes or annotators, which are difficult to obtain, especially in the medical domain. Self-supervised methods do not require explicit labels but do not always lead to the desired result. Discriminative methods would usually evaluate their phrase grounding performance by computing the cosine similarity between the text embeddings and the corresponding image embeddings. However, it has recently been shown that phrase grounding tasks can also be solved using generative models in an unsupervised context (Dombrowski et al., 2024). Specifically, text-to-image Latent Diffusion Models (LDMs) are useful, due to their use of cross-attention to combine the two modalities, as well as their ability to produce high-quality images. Text-to-image LDMs are trained to generate images from a dataset while receiving additional text conditioning from the corresponding text inputs (Dombrowski et al., 2023; Vilouras et al., 2024). Instead of using cosine similarity, the phrase grounding capabilities of LDMs are easier to evaluate by using their cross-attention layers. Earlier, Dombrowski et al. (2024) showed that using a frozen text encoder improves the phrase grounding capabilities of an LDM.

So far, the self-supervised approach by Boecking et al. (2022) achieved the highest phrase grounding performance on Chest X-ray (CXR) data by fine-tuning a Large Language Model (LLM) pre-trained on the biomedical domain on CXR reports. The resulting LLM is known as CXR-BERT. This model is jointly trained with an image encoder, in a framework called BioViL. In this work, we leverage these and use CXR-BERT as a frozen text encoder that conditions the U-Net in an LDM. Consequently, we inject the learned embeddings of CXR reports from CXR-BERT, while additionally fine-tuning the U-Net on corresponding CXR images. CXR-BERT and the LDM support each other in a bidirectional manner: the LDM, having a generative architecture, is able to leverage the full phrase grounding potential of the text embeddings compared to the simple CNN that is used in BioViL. Additionally, the powerful text embeddings learned by CXR-BERT provide the necessary conditioning to the LDM that enables the model to learn a well-grounded multimodal representation.

As a result, the contributions of our work include the following:

- We demonstrate that a multimodal text encoder with domain-specific knowledge can vastly improve phrase grounding capabilities of an LDM.

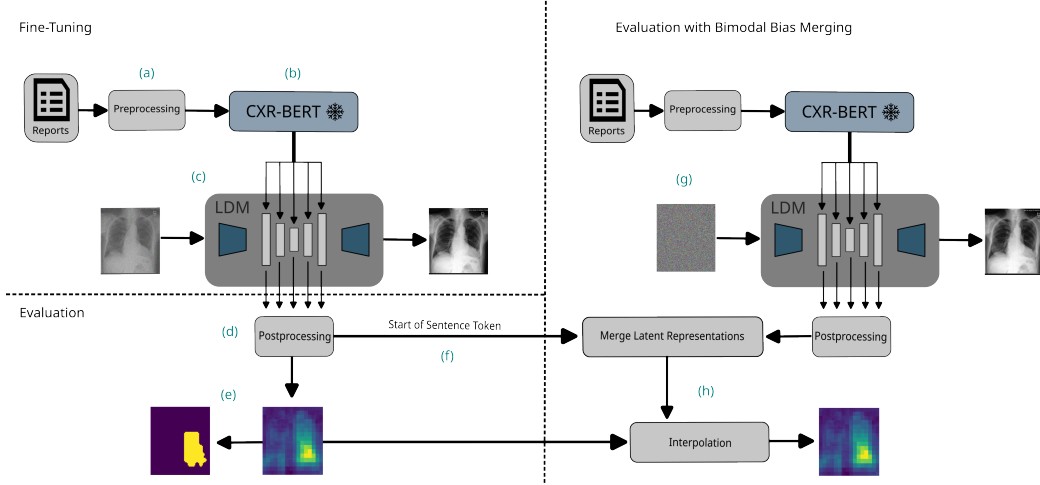

Figure 1: During fine-tuning, radiological text reports are extracted **(a)**. These reports are fed into CXR-BERT with frozen parameters **(b)**. The resulting text embeddings are used to condition the U-Net in the LDM, by injecting the embeddings into each cross-attention layer (represented as gray bars) in the U-Net. The LDM learns to generate images by getting noisy radiology images corresponding to the reports and a timestep as input **(c)**. During evaluation, noisy ground-truth images are repeatedly fed into the LDM to extract the corresponding cross-attention. These maps are processed based on relevant tokens and to get them into the correct dimensionality **(d)**. After a processing step, we obtain an activation map and its corresponding binary mask **(e)**. For BBM, we need to extract the image bias **(f)** and generate the text bias **(g)**, merge them and combine them with our original activation map **(h)**.

- We show that generative approaches can yield far better phrase grounding results than traditional discriminative approaches by nearly doubling conventual performance metrics such as mIoU.
- We discuss a novel post-processing method that can boost the phrase grounding capabilities of phrase grounding frameworks.

## 2. Method

### 2.1. Exchanging Text Encoders in Stable Diffusion

Our approach is based on fine-tuning Stable Diffusion (SD) (Rombach et al., 2022), a popular pre-trained text-to-image LDM. We only train the underlying U-Net of SD while keeping the text encoder frozen, since Dombrowski et al. (2024) demonstrated that maintaining the original configuration of the text encoder yields superior phrase grounding results.

We first fine-tune the U-Net by using the text encodings of the original pre-trained text encoder (CLIP-ViT-L/14) of SD version 1.5 (Rombach et al., 2022) on our training dataset.

Meanwhile, the parameters of the text encoder are kept frozen. We then compare these baseline results by replacing the original text encoder with a frozen CXR-BERT (Boecking et al., 2022) encoder in additional training runs. CXR-BERT is a multimodal language model pre-trained on the CXR domain. Due to being pre-trained on text and image inputs in a specific domain, CXR-BERT can provide better text representations than the standard text encoder of SDs, which is trained on a largely domain-agnostic subset of LAION-5B (Schuhmann et al., 2022).

Additionally, phrase grounding is inherently a task in which the image and the text modality need to be properly aligned. Therefore, LLMs that received both vision and textual learning signals are especially suited for phrase grounding tasks. This intuition is also supported by previous work, which already demonstrated that domain-specific text encoders with multimodal pre-training perform well during phrase grounding tasks (Boecking et al., 2022). Still, some research, such as the paper by Bluethgen et al. (2024) suggests that using a domain-specific text encoder for LDMs does not yield any benefits in the CXR domain. However, another interesting property of CXR-BERT (Boecking et al., 2022), is the use of both global and local loss. Typically, methods tend to use some variant of a global loss when pre-training a model, which computes the loss on image and phrase level. But including a local loss, that aligns words with image regions, better reflects the bottom-up structure of phrase grounding, since each individual token is associated with a region in the image.

## 2.2. Cross-Attention Map Extraction

An overview of the cross-attention map extraction process can be seen on the left side of Figure 1. The text inputs for CXR-BERT first need to be tokenized by its corresponding tokenizer with maximal token length $N_{\max}$ into tokens $\tau_1, \ldots, \tau_{N_{\max}}$. Since only words with lexical meanings can be mapped to image regions, as demonstrated in Figure 2, we remove tokens corresponding to function words by employing ScispaCy (Neumann et al., 2019). This approach yields a small improvement compared to the token processing method used by Dombrowski et al. (2024) (see Appx. B). From here, we are following the approach by Dombrowski et al. (2024), meaning we are mostly interested in the probability matrix $P$, defined as

$$P = \text{softmax}\left(\frac{QK^T}{\sqrt{d_k}}\right), \tag{1}$$

whereas $Q$ is the query, $K$ is the key and $d_k$ is the dimension of the attention embedding. In particular, for the batch size $B$, the layers $L$, the image height $H$ and the image width $W$, $K$ is a learned linear projection of the text embeddings with dimension $(B * L \times N_{\max} \times d_k)$, while $Q$ is a learned linear projection of the image embeddings with dimension $(B * L \times H * W \times d_k)$. Their inner product, the matrix $P$, is the basis for the cross-attention masks and is of dimension $(B * L \times H * W \times N_{\max})$. Consequently, for each sample in the batch and for each layer, P sets each pixel and each token embedding in relation to each other, making it suitable to evaluate visual grounding. This matrix is generated and saved for each timestep during inference. Therefore, when reshaping $P$ correctly and upsampling the image dimensions to our latent image size of 64, this results in a tensor with dimension $(B \times T \times L \times N_{\max} \times 64 \times 64)$ for the number of timesteps $T$. This allows us to easily select specific layers, timesteps and tokens. The 2D activation maps $P_{comb}$ can then be obtained

by simply averaging over the first three dimensions of each item in the batch and excluding the start and end tokens. This corresponds to computing the average over the attention maps for each timestep, layer and token. When we use lexical filtering, only relevant tokens are used for this averaging. An intuition for this approach is provided by Figure 2. Unlike other text encoders that would result in focusing on unnecessary details such as the ribcage for attention maps in higher resolutions, CXR-BERT produces strong attention maps across all resolutions in the U-Net, which is why we can average over the layers (see Appx. E for an example). Due to this consistency of CXR-BERT, we also do not need to select specific timesteps. To obtain accurate localization capabilities from these maps, the model needs both an image input and a textual input. Therefore, instead of starting with Gaussian noise during sampling, the ground-truth image is used as input in each timestep. The appropriate noise for the current timestep is added to a fresh input image in each step. The corresponding binary mask for computing mIoU is obtained via fitting a Gaussian Mixture Model to the activation map.

### 2.3. Bimodal Bias Merging

To further improve the accuracy of the 2D activation maps $P_{comb}$, we incorporate more information via a process we call Bimodal Bias Merging (BBM). An overview of this method can be seen on the right side of Figure 1. In this process, we combine the activation maps from Section 2.2 with the textual bias and image bias of the model, as motivated by the results of Section 4. To this end, we only extract the cross-attention values of $P$ that correspond to the start token, which represents the image bias of the model. This way, we end up with a tensor $P_{\text{img}}$ of dimension $(B \times T \times L \times 1 \times 64 \times 64)$. For the textual bias of the model, we need to sample the LDM again, but with the usual Gaussian noise as image input this time. This results in a tensor $P_{\text{txt}}$ of dimension $(B \times T \times L \times N_{\max} \times 64 \times 64)$. By combining $P_{\text{txt}}$ and $P_{\text{img}}$ via matrix multiplication (denoted as $\otimes$), we capture cross-modal interactions between the two representations. Empirically, $P_{\text{mult}} = P_{\text{img}} \otimes P_{\text{txt}}$ consists of large radial gradients that show the most likely locations of the disease. To have a measure for the accuracy of this map, we compute the structural similarity index measure (Wang et al., 2004) $s$ between the map for the text bias and the image bias, clipped to the range $[0, 1]$. Finally, to obtain our new activation map $P_{\text{BBM}}$, the bias interaction map and the original activation map are interpolated via the following quadratic Bézier curve:

$$P_{\text{BBM}} = 2(1-s)s\left(\frac{P_{\text{mult}} + P_{\text{comb}} + P_{\text{mult}} \odot P_{\text{comb}}}{2}\right) + (1-s)^2 P_{\text{comb}} + s^2 P_{\text{mult}} \quad (2)$$

with $\odot$ being the Hadamard product. This interpolation is essentially linear, except for the control point receiving additional information regarding the multiplicative interaction between the biases. In its base form, BBM is the linear interpolation $sP_{\text{comb}} + (1-s)P_{\text{mult}}$, which improves the activations around the location of the disease, as can be measured with CNR. However, this typically does not improve the generated attention maps, as measured with mIoU, since thresholding would include even low activations as part of the masks. For this purpose, we introduced a control point to the equation that essentially serves as a gating mechanism that constricts the activation areas to adhere to the merged biases. Therefore, it is primarily relevant when computing masks. By construction, Equation 2 remains close to

a linear interpolation, despite utilizing the gating mechanism, which considerably improves the activation maps, while also giving a slight boost to the masks. Since the activation maps are primarily supposed to increase interpretability, it should be easy to inspect them with the human eye. The main benefit of using BBM is that the activations are much clearer to see, which is more meaningful than simply using masks. For more details on the interpolation, see Appx. B.

In this way, the original map is combined with the modality interaction map based on the calculated confidence score. This is based on the heuristic that, if the image bias and text bias are similar, then the fields created by their merging are more likely to support finding an accurate location of the disease. Meanwhile, if the two biases have large discrepancies, their combined information is less likely to enhance the original activation map and should mostly be ignored.

## 3. Experiment Setup

### 3.1. Training Setup

Each run is performed using the same configuration: we use a base learning rate of $5e^{-05}$, with 1000 warmup steps and a cosine learning rate scheduler. The training is split over eight 80GB A100 GPUs with a batch size of 16 and two gradient accumulation steps each, resulting in an effective batch size of 256. To be reproducible, the used seeds were uniformly sampled and are 4200, 1759 and 6357. For more efficiency, the model weights are converted to mixed precision. During training, unconditional guidance training is applied, so the text conditioning would be dropped with a probability of 30%. Additionally, we keep an Exponential Moving Average of our U-Net, which is used for sampling.

### 3.2. Datasets

The base dataset for training and testing is the MIMIC-CXR dataset (Goldberger et al., 2000; Johnson et al., 2019). It contains pairings of CXR images and their respective reports, which include medical findings such as diseases. For training, we use the train split proposed by MIMIC-CXR-JPG (Goldberger et al., 2000; Johnson et al., 2024), which consists of 162,651 image-report pairs. Explorative hyperparamter optimizations were conducted on the ChestXRay14 dataset (Wang et al., 2017).

Our test set consists of MS-CXR (Boecking et al., 2022), a subset of MIMIC-CXR. MS-CXR features improved bounding boxes, which can be used to evaluate the phrase-grounding performance of our models. Additionally, MS-CXR includes refined report descriptions that yield higher evaluation accuracy.

### 3.3. Metrics

In order to be comparable with Boecking et al. (2022), we report the Contrast-to-Noise Ratio (CNR) and mean Intersection over Union (mIoU) of our results. CNR is calculated as $\text{CNR} = \frac{|\mu_{A_i} - \mu_{A_e}|}{\sqrt{\sigma_{A_i}^2 + \sigma_{A_e}^2}}$, where $\mu_{A_i}$ and $\mu_{A_e}$ represent the means and $\sigma_{A_i}^2$, $\sigma_{A_e}^2$ the variances of the similarity scores inside and outside the bounding box respectively. Therefore, CNR can be used to evaluate the phrase grounding performance without the need of applying

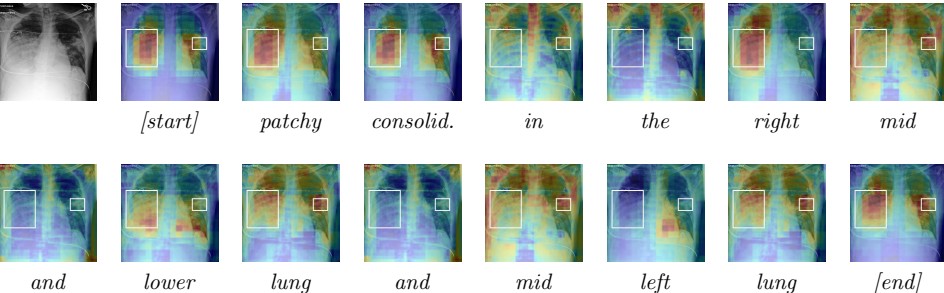

Figure 2: Shows an input CXR image in posterior-anterior view (so note that left and right are mirrored) together with the cross-attention maps for the tokens of the corresponding text report. Red shows high activations, while blue shows low activations. The white boxes indicate the ground-truth bounding boxes.

a threshold (Boecking et al., 2022). Additionally, we compute mIoU as the mean of the Jaccard distances $J(A, B) = \frac{|A \cap B|}{|A \cup B|}$ between the overlapping and non-overlapping regions of the thresholded phrase grounding image and the ground-truth bounding box.

## 4. Results & Discussion

In Figure 2, we can see how the model attends to different tokens in a sentence. For example, the token for *"consolidation"* shows the approximate region of the anomaly. The activations for *"right"* are on the right side, the one for *"lower"* is on the lower part and so on. Meanwhile, tokens with no lexical content, such as *"the"*, have no clear activation patterns. The activation map of the start token is the image bias of the model, which works already remarkably well. This implies that the model has a good internal representation of the diseases, even when considering the text and image modalities separately. Meanwhile, the end token provides even better phrase grounding capabilities, since it can incorporate the knowledge of the preceding tokens. Empirically, using the activation map of the end token produces similar results as using the mean of preceding tokens.

As shown in Table 4, using CXR-BERT for text-conditioning in an LDM leads to superior results in phrase grounding than using CXR-BERT in its original framework BioViL (Boecking et al., 2022). The cross-attention maps of the LDM yield better results for both metrics (*i.e.*, CNR and mIoU) reported by (Boecking et al., 2022). Our approach approximately doubles the mIoU results. Consequently, our approach seems to be especially suited for mask generation. Additionally, our setup achieves higher results across all diseases compared to BioViL, and for almost all diseases compared to its improved version, BioViL-L. As discussed in Appx. D, we can also confirm the observed trade-off between interpretability and image generation quality that was observed by Dombrowski et al. (2024). Our post-processing method BBM increases the CNR results considerably. The improvement for mIoU is smaller, which most likely stems from the fact that for the mask generation, the applied thresholding destroys some of the gained information. Notably, since our model could not learn pneumothorax properly, BBM decreases the phrase grounding performance for this disease. These low values for pneumothorax align with the

Table 1: Comparison of mIoU and CNR metrics on the MS-CXR dataset between CXR-BERT used as a text encoder for a LDM (CXR-BERT$_{LDM}$), in discriminative context (BioVil, adopted from Boecking et al. (2022), and with additional local loss (BioVil-L, adopted from Boecking et al. (2022)). Also includes results for our baseline using a frozen CLIP encoder (CLIP$_{LDM}$, reproduced from Dombrowski et al. (2024). Our results (CXR-BERT$_{LDM}$) are shown with BBM applied and without it, averaged over three training runs with different seeds (shown as mean±standard deviation). To be consistent with CLIP$_{LDM}$, we also used 50 timesteps during inference, although our method requires far less (see Appx. C).

| Disease | BioVil | | BioVil-L | | CLIP$_{LDM}$ | | CXR-BERT$_{LDM}$ | | +BBM | |
|---|---|---|---|---|---|---|---|---|---|---|
| | mIoU | CNR | mIoU | CNR | mIoU | CNR | mIoU | CNR | mIoU | CNR |
| Atelectasis | 0.296 | 1.02±.06 | 0.302 | 1.17±.04 | 0.425 | 1.14 | **0.58±.01** | 1.69±.02 | 0.55±.01 | **1.78±.01** |
| Cardiomegaly | 0.292 | 0.63±.08 | 0.375 | 0.95±.21 | 0.451 | 0.75 | 0.65±.00 | 1.37±.01 | **0.66±.01** | **1.54±.04** |
| Consolidation | 0.338 | 1.42±.02 | 0.346 | 1.45±.03 | 0.436 | 1.12 | **0.52±.00** | 1.62±.01 | 0.49±.00 | **1.68±.02** |
| Lung Opacity | 0.202 | 1.05±.06 | 0.209 | 1.19±.05 | 0.402 | 1.20 | **0.46±.00** | 1.57±.03 | 0.45±.00 | **1.67±.03** |
| Edema | 0.281 | 0.93±.03 | 0.275 | 0.96±.05 | 0.541 | 1.25 | **0.55±.01** | 1.35±.01 | 0.54±.01 | **1.37±.00** |
| Pneumonia | 0.323 | 1.27±.04 | 0.315 | 1.19±.01 | 0.438 | 1.12 | **0.53±.00** | 1.60±.01 | 0.51±.01 | **1.71±.01** |
| Pneumothorax | 0.109 | 0.48±.06 | 0.135 | **0.74±.05** | 0.312 | 0.22 | **0.34±.01** | 0.45±.09 | 0.33±.01 | 0.28±.07 |
| Pl. Effusion | 0.290 | 1.40±.06 | 0.315 | 1.50±.03 | 0.356 | 0.73 | **0.55±.00** | **1.65±.04** | 0.51±.01 | 1.62±.05 |
| Weighted Avg. | 0.266 | 1.03±.02 | 0.284 | 1.14±.04 | 0.409 | 0.72 | **0.54±.00** | 1.21±.03 | 0.53±.01 | **1.26±.04** |

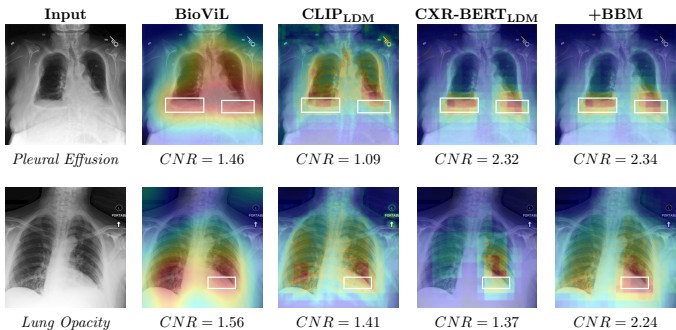

Figure 3: Examples for similarity maps of BioViL (results are reproduced from Boecking et al. (2022)), as well as the cross-attention maps of a LDM with a frozen CLIP text encoder (CLIP$_{LDM}$, reproduced from Dombrowski et al. (2024)), the LDM with a frozen CXR-BERT text encoder (CXR-BERT$_{LDM}$) and the same encoder with additional BBM post-processing (+BBM). Red means higher activations, while blue means lower activations. The ground-truth bounding boxes are given in white. The respective input images with their disease labels are on the left.

findings by Dombrowski et al. (2024), suggesting that this disease may be particularly challenging to model. This might be due to Pneumothorax being very inconsistent in terms of location and size, or the corresponding impressions being too short. Therefore, the model does not get enough information to model such a complex disease.

Furthermore, our approach improves the previous method by Dombrowski et al. (2024), which is also based on the extraction of cross-attention maps. However, the key difference is that we replaced the generic CLIP text encoder with the domain-specific CXR-BERT text encoder. This change greatly increases the performance of the model, as shown in Table 4. Consequently, we could show that a domain-specific LLM has the potential to greatly increase the phrase grounding potential of LDMs. These results demonstrate that better phrase grounding can be achieved in a generative context compared to a discriminative one, although the generative context has no specific alignment loss.

In Figure 3, one can observe examples of cross-attention maps extracted from the LDM conditioned with a CLIP text encoder and a CXR-BERT text encoder, as well as cosine similarity maps from BioViL. As already demonstrated in Dombrowski et al. (2024), employing a frozen CLIP text encoder yields solid phrase grounding results. However, it still performs worse than some domain-specific weakly supervised methods such as BioViL. By employing text conditioning based on an encoder with strong phrase grounding capabilities in that domain, the strengths of both methods are combined, resulting in the best outcomes. As we can see in the second row of Figure 3, BBM can sometimes correct inaccurate predictions made by our model, thus increasing its accuracy.

**Limitations:** First, while our model showed considerable improvements for the other diseases, it performed less optimally for Pneumothorax, which, as a disease, shows far less consistency in terms of shape and location in the body. Due to these characteristics, Pneumothorax is a challenging disease to localize for both our method and comparing methods. To properly learn to localize Pneumothorax, one example would be to directly fine-tune the model on a curated subset of the data, or alternatively, to take inspiration from other Pneumothorax detection works such as Park et al. (2022), and apply knowledge distillation from a model fine-tuned specifically on Pneumothorax. However, we kept our fine-tuning process as general as possible to enable a fair comparison between methods. Second, our BBM technique is strictly designed to complement models with strong phrase grounding capabilities and an LDM architecture. By focusing on well-aligned models, BBM optimally leverages their strengths, and the principles underlying our approach could inspire adaptations for other architectures or domains. A detailed ablation study is provided in Appx. A.

## 5. Conclusion

We demonstrated that domain-specific, multimodal text encoders, such as CXR-BERT, significantly enhance phrase grounding performance in LDMs, particularly in the medical imaging domain. By integrating such encoders, our approach nearly doubles key metrics like mIoU compared to state-of-the-art discriminative methods, establishing generative models as a superior alternative for this task. Additionally, we introduced BBM, which further refines cross-attention maps to improve localization accuracy and robustness.

Our findings highlight the untapped potential of generative models in aligning text and image modalities, providing a pathway toward more interpretable and trustworthy medical AI systems. While our work represents an advancement, it also underscores the importance of balancing interpretability with generative quality for clinical applications. Future research should focus on extending this approach to other medical domains and exploring strategies to optimize this balance further.

## 6. Acknowledgments

The authors gratefully acknowledge the scientific support and HPC resources provided by the Erlangen National High Performance Computing Center (NHR@FAU) of the Friedrich-Alexander-Universität Erlangen-Nürnberg (FAU) under the NHR projects b143dc and b180dc. NHR funding is provided by federal and Bavarian state authorities. NHR@FAU hardware is partially funded by the German Research Foundation (DFG) – 440719683.

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

# Appendix A. Ablation study

Table 2: Comparison of the average disease detection metrics across different sampling methods and fine-tuned LLMs using $timesteps = 50$. The CLIP results are reproduced from Dombrowski et al. (2024).

| Sampling Input | Model | mIoU | Top-1 | AUC-ROC | CNR |
|---|---|---|---|---|---|
| Text-Conditioning | CLIP (Rombach et al., 2022) | 0.3540 | 0.1694 | 0.6304 | 0.3571 |
| | OpenCLIP (Ilharco et al., 2021) | 0.3337 | 0.0828 | 0.5676 | 0.1634 |
| | RadBERT (Chambon et al., 2022) | 0.3535 | 0.2929 | 0.6693 | 0.4901 |
| | Med-KEBERT (Zhang et al., 2023) | 0.3188 | 0.1400 | 0.5715 | 0.2011 |
| | CXR-CLIP (You et al., 2023) | 0.3556 | 0.2477 | 0.6504 | 0.4319 |
| | CXR-BERT (Boecking et al., 2022) | 0.2932 | 0.0694 | 0.5895 | 0.2343 |
| CFG | CLIP (Rombach et al., 2022) | 0.3838 | 0.2422 | 0.6606 | 0.4403 |
| | OpenCLIP (Ilharco et al., 2021) | 0.3226 | 0.0443 | 0.5512 | 0.1195 |
| | RadBERT (Chambon et al., 2022) | 0.3542 | 0.2709 | 0.6604 | 0.4558 |
| | Med-KEBERT (Zhang et al., 2023) | 0.3190 | 0.2134 | 0.5880 | 0.2496 |
| | CXR-CLIP (You et al., 2023) | 0.3556 | 0.2477 | 0.6504 | 0.4319 |
| | CXR-BERT (Boecking et al., 2022) | 0.4541 | 0.4230 | 0.7530 | 0.7730 |
| GT-1 + CFG | CLIP (Rombach et al., 2022) | 0.3528 | 0.2357 | 0.6607 | 0.4568 |
| | OpenCLIP (Ilharco et al., 2021) | 0.3302 | 0.0435 | 0.5873 | 0.2245 |
| | RadBERT (Chambon et al., 2022) | 0.3162 | 0.2983 | 0.6894 | 0.5644 |
| | Med-KEBERT (Zhang et al., 2023) | 0.3336 | 0.2900 | 0.6367 | 0.3897 |
| | CXR-CLIP (You et al., 2023) | 0.3588 | 0.2589 | 0.6669 | 0.4810 |
| | CXR-BERT (Boecking et al., 2022) | 0.4067 | 0.39823 | 0.8060 | 0.9886 |
| GT + CFG | CLIP (Rombach et al., 2022) | 0.4089 | 0.4062 | 0.7413 | 0.7164 |
| | OpenCLIP (Ilharco et al., 2021) | 0.3344 | 0.0534 | 0.6451 | 0.4041 |
| | RadBERT (Chambon et al., 2022) | 0.3995 | 0.4138 | 0.7781 | 0.8363 |
| | Med-KEBERT (Zhang et al., 2023) | 0.3452 | 0.3508 | 0.6893 | 0.5536 |
| | CXR-CLIP (You et al., 2023) | 0.3684 | 0.2695 | 0.6979 | 0.5809 |
| | CXR-BERT (Boecking et al., 2022) | 0.5242 | 0.6506 | 0.8657 | 1.2288 |

The ablation studies investigate the impact of different domain-specific text encoders on phrase grounding. As shown in Table 2, four different sampling methods are compared: only using generic text conditioning during sampling, applying Conditional Free Guidance (CFG) during sampling, additionally giving the noisy ground-truth image in the first timestep (GT-1 + CFG) and additionally giving the noisy ground-truth image in every step (GT + CFG). The metrics AUC-ROC and Top-1 (Dombrowski et al., 2024) are also included here. The most important detail that can be seen in Table 2, is that CXR-BERT performs by far the best of all tested models. There are several attributes that distinguish CXR-BERT from the rest that could be responsible for that. Unlike the domain-agnostic CLIP models, CXR-BERT is trained on domain specific CXR data. In contrast to RadBERT, CXR-BERT is trained in a multimodal manner. Compared to Med-KEBERT, CXR-BERT does not rely on any report preprocessing. In comparison to CXR-CLIP, CXR-BERT has a considerably more complex pretraining procedure. Additionally, CXR-BERT uses both local and global

loss, which differentiates it from all other discussed models. Adding a local loss term, paired with the domain-specific, multimodal training is most likely the key to the strong performance of CXR-BERT.

## Appendix B. Processing Methods Comparison

Table 3: Comparison of different processing methods on CXR-BERT$_{\text{LDM}}$ for our primary phrase grounding metrics. Disease filtering refers to the token processing method used by Dombrowski et al. (2024), while lexical filtering refers to our token processing method that filters words without lexical meaning. Linear Bézier refers to the linear interpolation of the image and text bias. The quadratic Bézier approach uses the multiplicative interaction of the biases as control point. Mixture Bézier blends the quadratic and linear approach.

| Disease | Disease Filtering | | Lexical Filtering | | Linear Bézier | | Quadratic Bézier | | Mixture Bézier | |
|---|---|---|---|---|---|---|---|---|---|---|
| | mIoU | CNR | mIoU | CNR | mIoU | CNR | mIoU | CNR | mIoU | CNR |
| Atelectasis | 0.506 | 1.53 | 0.589 | 1.71 | 0.548 | 1.79 | 0.576 | 1.78 | 0.559 | 1.79 |
| Cardiomegaly | 0.649 | 1.34 | 0.649 | 1.38 | 0.657 | 1.57 | 0.685 | 1.54 | 0.670 | 1.58 |
| Consolidation | 0.522 | 1.48 | 0.520 | 1.63 | 0.488 | 1.68 | 0.518 | 1.69 | 0.498 | 1.69 |
| Lung Opacity | 0.418 | 1.35 | 0.462 | 1.55 | 0.438 | 1.63 | 0.470 | 1.67 | 0.451 | 1.66 |
| Edema | 0.556 | 1.43 | 0.543 | 1.34 | 0.556 | 1.38 | 0.543 | 1.36 | 0.534 | 1.38 |
| Pneumonia | 0.510 | 1.48 | 0.536 | 1.59 | 0.508 | 1.70 | 0.539 | 1.68 | 0.516 | 1.70 |
| Pneumothorax | 0.308 | 0.71 | 0.336 | 0.55 | 0.318 | 0.41 | 0.356 | 0.36 | 0.332 | 0.37 |
| Pl. Effusion | 0.565 | 1.69 | 0.551 | 1.69 | 0.565 | 1.69 | 0.535 | 1.67 | 0.514 | 1.69 |
| Weighted Avg. | 0.524 | 1.23 | 0.537 | 1.24 | 0.526 | 1.31 | 0.558 | 1.28 | 0.539 | 1.31 |

In Table 3, we can see how different processing methods affect the phrase grounding performance of our model. The first two columns are concerned with which tokens should be considered for the creation of the activation maps. Neither start nor end tokens are considered in either approach. Dombrowski et al. (2024) only used the tokens corresponding to the disease if at least one is present. Otherwise, if the disease is not mentioned in the report, all tokens are used. Meanwhile, our approach filters any words with no lexical meaning, and then uses the remaining tokens. The approach is motivated by the findings seen in Figure 2. This change yields a small improvement in phrase grounding performance compared to the original method.

The remaining three columns of Table 3 are concerned with different interpolation techniques that can be used for BBM. Linear Bézier refers to a usual linear interpolation, meaning the equation

$$sP_{\text{mult}} + (1 - s)P_{\text{comb}} \tag{3}$$

is used. While this method improves CNR considerably, it is not well-suited for mask generation, since the larger activation areas lead to masks that are too large. To fix this issue, Quadratic Bézier incorporates a quadratic Bézier curve for interpolation, namely

$$2(1 - s)s(P_{\text{mult}} \odot P_{\text{comb}}) + (1 - s)^2 P_{\text{comb}} + s^2 P_{\text{mult}}. \tag{4}$$

The control point of the interpolation is the Hadamard product of $P_{\text{mult}}$ and $P_{\text{comb}}$. This serves primarily two purposes: first, to catch more complex multiplicative interactions between the two matrices. Second, the multiplication with $P_{\text{mult}}$ acts as a gating mechanism, which means using the multiplicative interaction as the midpoint hinders the interpolation from having overly large areas of activation. Instead, the interpolation has a greater focus on the relevant areas of activation. Using this approach, the mask generation can be successfully improved, but at the cost of a lower improvement of CNR. Consequently, we blend the linear and quadratic approach to gain the benefits of both, as can be seen in the Mixture Bézier column. The result is an essentially linear interpolation which midpoint is combined with $P_{\text{mult}} \odot P_{\text{comb}}$. Since a linear Bézier curve can be expressed as a quadratic Bézier curve with

$$2(1-s)s \left( \frac{P_{\text{mult}} + P_{\text{comb}}}{2} \right) + (1-s)^2 P_{\text{comb}} + s^2 P_{\text{mult}}, \tag{5}$$

incorporating the Hadamard product results in

$$2(1-s)s \left( \frac{P_{\text{mult}} + P_{\text{comb}} + P_{\text{mult}} \odot P_{\text{comb}}}{2} \right) + (1-s)^2 P_{\text{comb}} + s^2 P_{\text{mult}}. \tag{6}$$

This interpolation achieves a balance between the accuracy of the activation maps and their corresponding masks.

## Appendix C. Hyperparameter Optimization

All hyperparemter optimization was carried out on the ChestXRay14 (Wang et al., 2017) dataset. In addition to CNR and mIoU, we also include the Top-1 metric (Dombrowski et al., 2024) and the AUC-ROC metric here. Figure 4 demonstrates that after about five timesteps, there are no significant improvements in phrase grounding performance during image generation using a noisy ground-truth image as input. This indicates that our results do not depend heavily on this hyperparameter. It seems reasonable that the results saturate early for this sampling method compared to the others, since the model is given significantly more information in form of the ground-truth images. Using the noisy ground-truth image only in the first denoising step is the same to our main approach initially. This behavior is intuitive, since using the ground-truth only in the first step means essentially using it in all steps if only a single timestep is used in total. As the number of timesteps increases, the performance gradually becomes closer to the results from text-only conditioning. However, while there is a steep decline after the first few steps if the ground-truth image is only used in the first sampling step, the performance eventually stabilizes.

Figure 5 showcases how the selection of the last $n$ timesteps during ground-truth sampling affects the results. The figure is constrained to 65 timesteps, since this configuration produced the best results given all of the four phrase grounding metrics. Also, this can be seen as an exemplary result, since the observed trends are very similar for all timesteps in a sensible range. As can be seen in Figure 5, the metrics do not show the same behavior over the number of selected timesteps. CNR and AUC-ROC steadily increase the more timesteps are selected. However, the progression of mIoU is concave, peaking at selecting

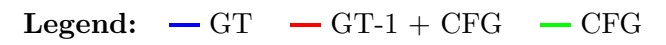

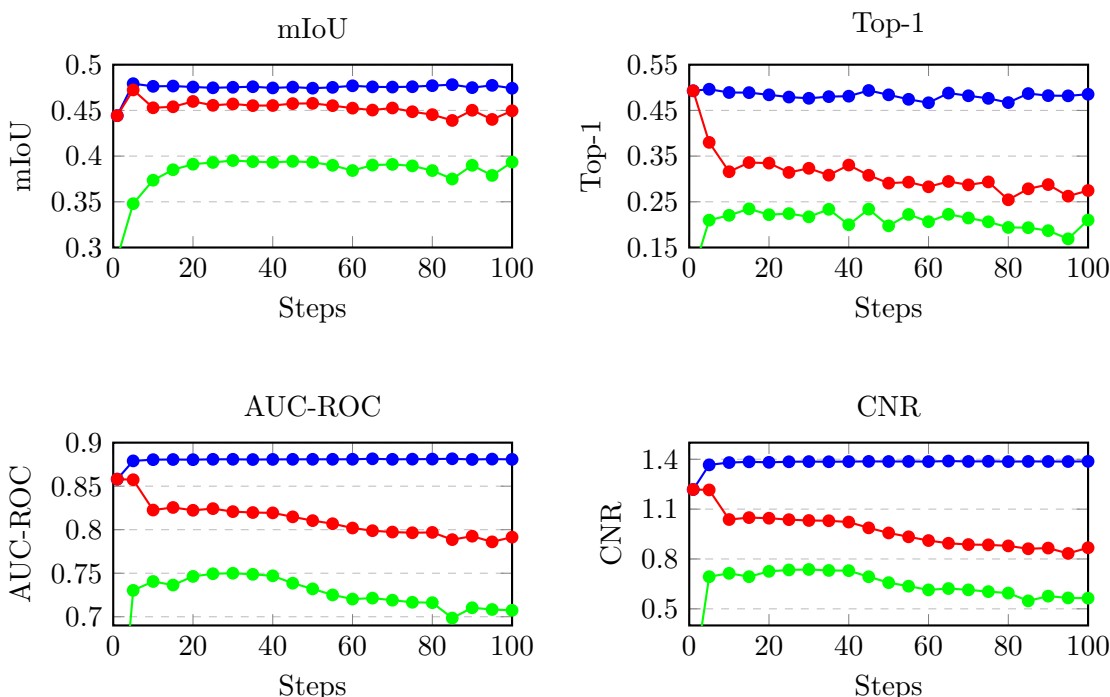

Figure 4: Performance of CXR-BERT using different sampling modes, trained on the MIMIC dataset, across various timesteps when evaluated on the ChestXRay14 dataset. The shown sampling modes are CFG with the ground truth image in the initial denoising step (GT-1 + CFG) and CFG with the ground truth image in every denoising step (GT).

45 timesteps and then decreasing. Meanwhile, Top-1 peaks at 5 timesteps and then shows a tendency to increase over time.

Since AUC-ROC and CNR are closely related metrics, it makes sense that they again show similar behavior. When incorporating more timesteps, noise introduced in single timesteps becomes less relevant. Both AUC-ROC and CNR give worse results when more noise is introduced to the signal, which is why these metrics typically perform better for a larger number of timesteps. Meanwhile, Top-1 only incorporates the highest activations, which is why this metric is extremely robust to noise. Therefore, only selecting a low number of timesteps can work well. Top-1 most likely decreases when selecting a larger number of timesteps, since early timesteps focus on coarse features, resulting in larger activation areas. Consequently, it is more likely that the highest activation is no longer strictly within the ground-truth bounding box. Meanwhile, the mIoU metric can tolerate a certain amount

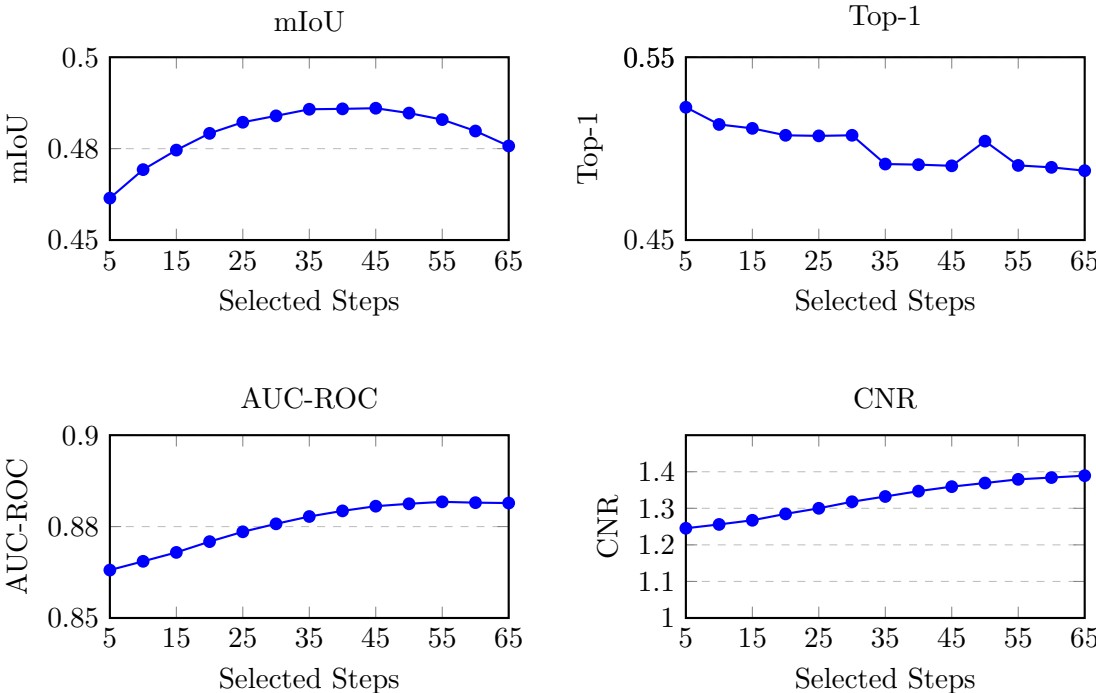

Figure 5: Results for LDM ground-truth sampling conditioned by CXR-BERT on the MIMIC-CXR dataset.

of noise, due to the thresholding applied when creating the binary masks. However, our masking approach generally has a tendency to include too much of the signal as part of the mask. So both too much noise and too large activation areas decrease the quality of the generated mask. Therefore, a compromise between both, resulting in a selection of roughly half of the timesteps, results in the best mIoU values in this setup.

When looking at the results, one should keep in mind that the changes are all very low, so the number of selected timesteps does not play a considerable role as a hyperparameter either.

## Appendix D. Interpretability Trade-off

While the presented phrase grounding performance of our model allows for a higher degree of interpretability than comparable models, this comes at a price. Dombrowski et al. (2024) discuss a trade-off between interpretability and performance in LDMs. They note that models with weaker phrase grounding capabilities often produce lower-quality images, while those with higher image quality tend to have poorer phrase grounding performance. Table 4 shows that this pattern is evident in our experiments as well. Using a frozen CXR-BERT encoder for text conditioning results in lower FID scores, but produces the best phrase grounding performance. Meanwhile, the frozen CLIP encoder, which has weaker phrase grounding, achieves better FID scores. A learnable CLIP encoder provides the highest image

Table 4: Comparison of FID and $FID_{XRV}$ values for images generated by LDMs using different text encoders. CNR values are also included to highlight the negative correlation between CNR and FID. The frozen CLIP results is reproduced and the learnable CLIP result is adopted from Dombrowski et al. (2024).

| Model | FID | $FID_{XRV}$ | CNR |
|---|---|---|---|
| Frozen CXR-BERT | 109.4 | 18.0 | 0.84 |
| Frozen CLIP | 83.8 | 14.5 | 0.72 |
| Learnable CLIP | 61.9 | 7.7 | 0.13 |

quality, but the lowest phrase grounding metrics. This means that choosing the correct model for application in the clinical field needs to be carefully considered. Simply using the model with the best image generation capabilities might produce good images. However, no trust can be put into the fidelity of these images, since their internal representations cannot be interpreted. Additionally, their lacking alignment between the image and text modalities imply that these models have no proper understanding of what they are generating, which might lead to harmful biases and mistakes in the generated images. Models with better phrase grounding capabilities might be more trustworthy, but their generated images lacking in quality can also be problematic for clinical applications. Even if the model has a good internal representation of the modalities, if the model generates subpar or unrealistic images, these can hardly be used in clinical settings. Currently, professionals need to choose a fitting balance between between quality and interpretability depending on their use case.

## Appendix E.  Attention Maps on Different Resolutions

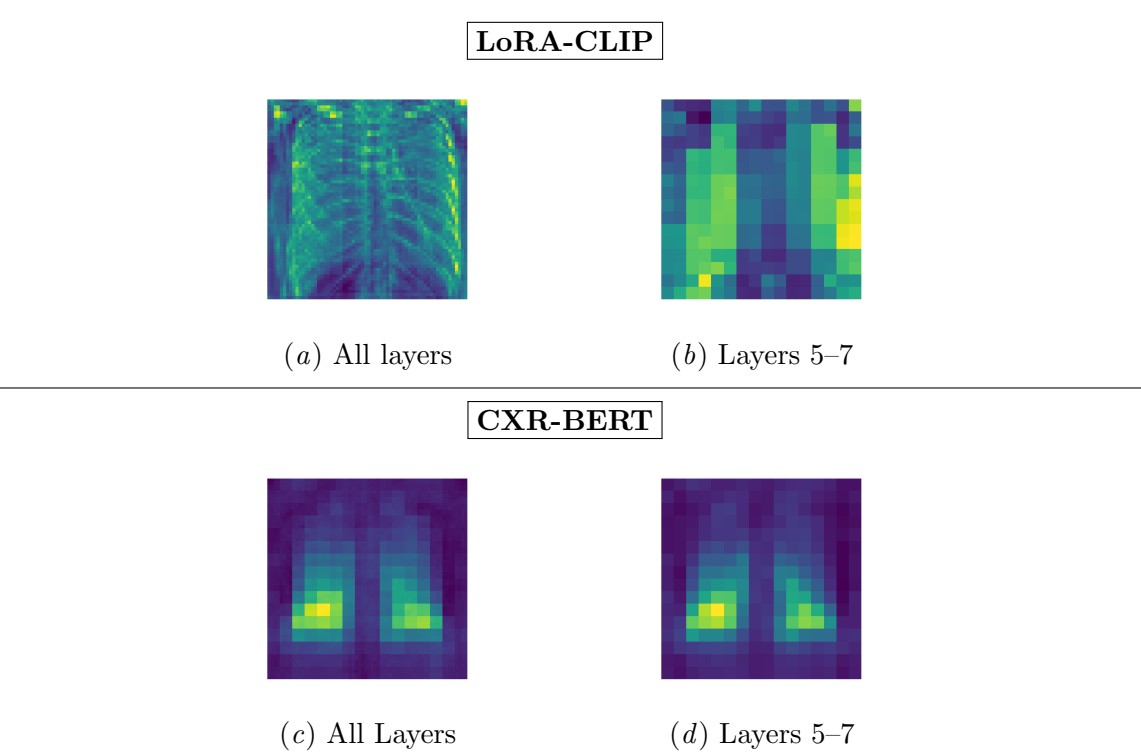

Figure 6: Comparison of cross-attention maps between conditioning provided by a CLIP model fine-tuned via LoRA (Hu et al., 2022) and one conditioned by CXR-BERT. It can be seen that LoRA provides no localization capabilities and can consequently profit from the lower resolution through selecting the middle layers, while this does not work on CXR-BERT.

As can be seen in Figure 6, models that have not learned a proper alignment between the image and text modalities can benefit from only selecting the innermost attention layers (*i.e.*, the layers with the lowest resolution. However, this is mainly due to lower resolutions naturally being closer to an activation area compared to fine-grained features. Inspecting all layers highlights that, in truth, such models have not learned a proper alignment between the two modalities and instead focus on unnecessary details such as the ribcage. In contrast, CXR-BERT has learned a proper alignment, so we can simply average over all layers to get our results.

