# OpenReview forum: "Generate to Ground: Multimodal Text Conditioning Boosts Phrase Grounding in Medical Vision-Language Models"
_MIDL.io/2025/Conference — MIDL 2025 Poster_

### Official Review · Reviewer_Q4qg · 2025-02-18

**Confidence:** 3
**Preliminary Rating:** 4
**Recommendation:** Poster

**Summary:**

This paper introduces a generative approach for phrase grounding in medical imaging, leveraging Latent Diffusion Models (LDMs) and cross-attention maps to enable grounding without explicit bounding box annotations. The model integrates CXR-BERT as a frozen multimodal encoder, injecting structured text embeddings into the U-Net of LDM, enhancing text-image interaction. Additionally, a Bimodal Bias Merging (BBM) technique is proposed to refine cross-attention maps. Experimental results on chest X-ray phrase grounding show superior mIoU scores over discriminative methods.

**Strengths:**

1. This paper introduces a generative approach for phrase grounding, which is an interesting and innovative alternative to traditional discriminative methods, potentially offering better flexibility and generalization.
2. The study demonstrates that incorporating a multimodal text encoder enriched with domain-specific knowledge can enhance the phrase grounding capabilities of a Latent Diffusion Model.
3. The topic of phrase grounding is highly significant and has practical implications for the medical AI community, particularly in improving interpretability and clinical decision support.
4. The paper is well-structured and clearly written, making it easy to follow and understand.

**Weaknesses:**

1. The proposed BBM module provides limited improvements in grounding performance. The Contrast-to-Noise Ratio is not a robust metric for evaluating phrase grounding, and the mIoU scores show no improvement after adding BBM, with six out of eight tasks experiencing a performance drop in Table 1.
2. The paper mentions that “our model could not learn pneumothorax properly”, but does not provide a clear explanation for this limitation. Why does the model fail on pneumothorax?
3. The effectiveness of the proposed model should be further validated on more diverse datasets (like segmentation) to demonstrate its generalizability beyond the current experimental setting.

**Detailed Comments:**

The paper would benefit from a more in-depth discussion on visual grounding in the medical domain, such as [1, 2, 3], providing context on how the proposed generative approach compares to existing visual grounding techniques.

[1] Ichinose A, Hatsutani T, Nakamura K, et al. Visual grounding of whole radiology reports for 3d ct images[C]//International Conference on Medical Image Computing and Computer-Assisted Intervention. Cham: Springer Nature Switzerland, 2023: 611-621.
[2] Chen Z, Zhou Y, Tran A, et al. Medical phrase grounding with region-phrase context contrastive alignment[C]//International Conference on Medical Image Computing and Computer-Assisted Intervention. Cham: Springer Nature Switzerland, 2023: 371-381.
[3] Zou K, Bai Y, Chen Z, et al. MedRG: Medical Report Grounding with Multi-modal Large Language Model[J]. arXiv preprint arXiv:2404.06798, 2024.

**Justification Of The Preliminary Rating:**

This paper introduces a generative approach for phrase grounding in medical imaging, leveraging LDMs and CXR-BERT to enhance grounding accuracy without requiring explicit bounding box annotations. The proposed BBM further refines localization accuracy. Experimental results show strong improvements in mIoU scores over traditional discriminative models. However, BBM’s effectiveness is limited, with performance declines in several tasks.
Overall, the novel generative approach is promising, and despite some methodological and evaluation limitations, the paper makes a valuable contribution to the field. I lean towards acceptance, with minor refinements needed to improve clarity and generalization.

**Questions To Address In The Rebuttal:**

Please refer to Weaknesses and Comments

**Special Issue:**

Yes

---

> ### Author Response · Authors · 2025-03-07
>
> ### **Weakness 1:** CNR is an Unsuitable Metric to Evaluate Visual Grounding
>
> We believe that CNR is an important metric in our context, since unlike mIoU, it is not dependent on a threshold to compute masks, but instead evaluates the activation maps we want to investigate directly.
> This is especially relevant since our approach is supposed to increase the interpretability of biomedical vision-language models.
> In contrast to attention masks, attention maps are able to capture a more detailed insight into the alignment of the modalities in the model.
> Methods that increase the quality of such activation maps to the human eye, such as the linear variants of our BBM method, are valuable for interpreting the alignment results in more detail. To achieve this, we need metrics such as CNR and cannot solely rely on metrics such as mIoU.
>
> ### **Weakness 2:** Challenges with Pneumothorax
>
> The challenge with Pneumothorax is, that it shows no consistency in both location and size.
> Furthermore, textual descriptions of the disease are typically rather limited in the provided impressions.
> So from the textual side, only knowing that the patient has Pneumothorax gives the model very limited information regarding the position of the disease, which makes it very hard to learn.
> This problem is not unique to our approach, both main references of our paper, *i.e.*, Dombrowski et al.  (2024) and Boecking et al. (2022), show the weakest performance on Pneumothorax.
>
> **Additionally, we added some suggestions for improving the performance for Pneumothorax to our manuscript.**
>
> ### **Weakness 3:** Further Validation on More Diverse Datasets Needed
>
> The data-intensive nature of phrase grounding makes it challenging to evaluate methods across multiple modalities and datasets. While our experiments on two datasets (MIMIC-CXR and ChestXRay-14) provide substantial evidence that our method generalizes well, we plan to extend our work to other modalities. Notably, applying text-to-image models requires large datasets paired with radiology reports and segmentation masks, necessitating careful manual curation. One potential approach to addressing this challenge is leveraging VLMs for automated report generation. However, this would require domain expertise and rigorous validation to ensure the generated reports are meaningful, which is beyond the scope of this paper.
>
> ### Detailed Comment:
>
> **We added a more thorough discussion of visual grounding in medicine in Section 1.**

---

### Official Review · Reviewer_tyMn · 2025-02-20

**Confidence:** 4
**Preliminary Rating:** 2

**Summary:**

This work claims that using domain specific text encoder like CXR-BERT in phrase grounding can boost the performance in mIoU especially. In addition, with the proposed post processing mechnism called BBM, the performance is further advanced.

**Strengths:**

This work demonstrates substantial improvements across eight diseases, achieving significant gains in both mIoU and CNR metrics. The proposed generative approach, combined with Bimodal Bias Merging, effectively refines cross-attention maps, leading to superior phrase grounding performance.

**Weaknesses:**

The primary weakness of this work lies in the presentation of Section 2. Neither Figure 1 nor the textual description clearly convey how the proposed framework functions. Even after multiple readings, the training process of the model remains unclear. Below are specific concerns that need to be addressed:

1. In figure 1, the meaning of the arrows and gray bars within the LDM boxes is ambiguous. Do the arrows represent addition, multiplication, or concatenation? Does the gray bar indicate hidden layers within the U-Net?

2. The output of CXR-BERT is a text embedding. How is this vector injected into the latent layers of the U-Net, assuming the gray bar represents those layers?

3. The role of text embeddings in the training process of the diffusion model is not clearly explained. A more explicit description would be helpful.

4. Section 2.2 introduces the probability matrix P for generating the activation map P_BBM. However, since P is purely an attention weight matrix, it does not inherently carry visual meaning. Why does averaging across the first three dimensions of the attention weight result in an activation map?

5. If I understand correctly, the query and key representations are derived from the frozen CXR-BERT. From which layer are Q and K extracted?

6. How is s in eq 2 determined? Is the reported performance dependent on a carefully tuned s, or is the method robust across different values?

7. The work does not sufficiently justify the design choices of each component. A deeper insight into why each part of the framework was structured in a particular way would strengthen the contribution. Given MIDL’s standard for rigorous methodological contributions, this aspect is lacking.

8. The primary innovation over Dombrowski et al. (2024) appears to be the introduction of CXR-BERT and BBM. However, it is unclear whether BBM is effective under a wide range of parameter choices or if it only works under specific settings. Without further clarification on BBM’s robustness, the contribution of this work remains marginal.

**Detailed Comments:**

Author argues that providing standard deviation to the experiment result would requires substantial computational resources. I would suggest author use adjacent checkpoints' result as a surrogate, which can provide the reader a more clear view on the robustness of the method.

**Justification Of The Preliminary Rating:**

While the proposed approach demonstrates substantial improvements on benchmark metrics, the current presentation of the work does not meet the standard expected for publication at a venue like MIDL. The lack of clarity in key methodological details, particularly regarding model training, text embedding integration, and the interpretation of attention-based activation maps, limits the accessibility and impact of the work. Addressing these concerns would significantly enhance the scientific value and rigor of the study, making it a stronger contribution to the community.

**Questions To Address In The Rebuttal:**

Please address my question in weakness

**Special Issue:**

No

---

> ### Author Response · Authors · 2025-03-07
>
> ### **Weaknesses 1,2,3,5,7:** Unclear Description and Motivation of our Method
>
> The grey bars indeed indicate hidden layers of the U-Net, in particular the cross-attention layers of it. As described in the original LDM paper by Rombach et al. (2022), the underlying U-Net uses additional cross-attention layers, which enables the combination of the image and text modalities. This mechanism allows for text-conditioning in diffusion models. Likewise, the arrows indicate that the text embedding is fed into each cross-attention layer of varying resolution.
>
> Therefore, the role of the text embeddings for diffusion models in general, is that they provide conditioning to the probability path which guides the image generation process.
> We use the fact that the text embeddings are incorporated via cross-attention, which allows us to inspect the visual alignment between the two modalities.
>
> The cross-attention computation basically works as described in the original transformer, *i.e.*, the text embeddings are transformed into the so-called key K via a learned linear projection and the corresponding  image embedding is transformed into the query Q via another learned linear projection. Q has dimensions (B ∗ L × H ∗ W × D)  for batch size B, layers L, image height H, image width W and projection dimensionality D.
> K has dimensions (B ∗ L × N × D) for the maximal token length N.
> The inner product resulting in the probability matrix P has dimensions (B ∗ L × H ∗ W × N). Consequently, for each sample in the batch and for each layer, P sets each pixel and each token embedding in relation to each other, making it suitable to  evaluate visual grounding.
>
> **We have extended Section 2.2 to give a more in-depth description of this process. Also, we have extended the description of Figure 1 to make it clearer.**
>
> ### **Weakness 4:** Reasoning Behind Computing the Activation Maps via Simple Averages
>
> The averaging across the first layers of P works due to careful reshaping of the matrix beforehand and since all layers of a LDM are spatially coherent (e.g. no flattening layer exists).
> As mentioned above, matrix P originally has dimensions (B ∗ L × H ∗ W × N) for batch size B, layers L, image height H, image width W and token length N.
> First, we want to split the products in the first two dimensions into their respective terms.
> During sampling, such a tensor is computed for each timestep during inference, which yields a tensor of dimensions (L × T × B × N ×  H × W ) for the number of timesteps T.
> The image dimensions for each U-Net layer is different.
> Therefore, we upsample W and H to be always 64, which is the dimension of the latent images used as input to the U-Net.
> Finally, we just need to rearrange the layers to get our desired tensor of dimension (B × T × L × N × 64 × 64).
> So for a given attention representation in the batch, we average over all timesteps, resolutions and selected tokens to get attention images of dimension (64,64).
>
> Why averaging over selected tokens is meaningful can be seen in Figure 2 of our paper.
> Regarding the intuition behind averaging over the layers and timesteps, please refer to the updated manuscript.
>
> **We extended Section 2.2 of our paper to contain more relevant details concerning the attention map extraction.
> Also, we added Appendix E to give an intuition why one can average over the layers using CXR-BERT, while it might not work using other text embeddings.**
>
> ### **Weaknesses 6,8:** Unclear Aspects of BBM
>
> We did not apply any fine-tuning tothe parameter s, which controls the interpolation between the merged bias tensor and the original bimodal tensor.
> As mentioned in Section 2.3, the parameter s is computed as the structural similarity index measure between the image bias map and the text bias map, clipped to the range \[0,1\].
> We chose this similarity measure, since the two maps are essentially images and the structural similarity is a useful metric to capture their differences.
> The reason for using a similarity measure between the two bias images in the first place is straightforward: if the image bias and text bias are different, the model would locate the disease differently given only the image or only the textual description respectively. This is an indicator that the model is unsure about either the disease image or description, which means that the merging of these biases is unlikely to provide meaningful results.
>
>  The essential idea for Equation 2 is, that a linear interpolation strengthens activations around the disease localization, which is great for interpretability when inspecting activation maps, but is not necessary useful for mask computation, so we constructed an essentially linear interpolation with an additional gating mechanism to restrain masks to relevant areas *(more details in manuscript)*.
>
> **We extended Section 2.3 to include more details on the reasoning behind Bimodal Bias Merging.**
>
> ### Detailed Comment:
>
> **We added more training runs to our main results.**

---

> ### Comment · Area_Chair_exgM · 2025-03-21
>
> Dear reviewer,
>
> In this case the paper has highly varied scores that average to a borderline opinion. GIven the authors's rebuttal, is your score unchanged or do you have any update?

---

### Official Review · Reviewer_eypD · 2025-02-22

**Confidence:** 3
**Preliminary Rating:** 4
**Recommendation:** Poster

**Summary:**

This paper proposes a generative approach to enhance phrase grounding tasks in medical vision-language models. By combining a domain-specific text encoder (like CXR-BERT) with a text-to-image diffusion model framework, the paper demonstrates significant advantages in zero-shot phrase grounding.

**Strengths:**

1. The combination of text-to-image diffusion models and domain-specific text encoders achieves significant improvements in phrase grounding tasks.
2. The method nearly doubles existing state-of-the-art results on key metrics like mIoU and CNR, demonstrating its effectiveness.
3. The Bimodal Bias Merging technique enhances localization accuracy by aligning text and image biases.

**Weaknesses:**

1. Training and inference with diffusion models require substantial computational resources
2. The model shows weaker performance on diseases like pneumothorax, suggesting limitations in handling anomalies with high variability in location and shape.

**Detailed Comments:**

The authors introduce a novel post-processing technique—Bimodal Bias Merging (BBM)—which aligns text and image biases to further improve localization accuracy. Experimental results show that this method outperforms state-of-the-art discriminative approaches in the medical imaging domain, nearly doubling performance on key metrics like mIoU and CNR.

**Justification Of The Preliminary Rating:**

The paper uses generative mdoel for phrase grounding in medical image and provide some performance improvements. The are some concerns but overall the paper is well-supported by the experiment result. The contribution is good and the topic is suitable for this conference.

**Questions To Address In The Rebuttal:**

1. Can the author provide insights that if it is possible to actually deploy it in real word clinic given the computational resource limited situation?
2. Is the method can be extended to improve performance on more challenging disease categories like pneumothorax?

---

> ### Author Response · Authors · 2025-03-07
>
> ### **Weakness 1:** Computational Resource Demand of LDMs
>
> While diffusion models indeed require significant computational resources, our method mitigates one of the most important aspects in this regard, namely the number of inference steps needed.
> As shown in Appendix C of our paper, our method considerably outperforms competing state-of-the-art methods, even if we only use a single timestep for inference.
> Consequently, compared to many other diffusion-based methods, our method needs far less resources during inference, which is essential for clinical deployment.
>
>  **We added a reference to the required number of inference steps to our manuscript.**
>
>
> ### **Weakness 2:** Need for Possible Improvements Regarding Challenging Diseases such as Pneumothorax
>
> One straightforward way that should improve the performance on more challenging disease like Pneumothorax, is fine-tuning on such diseases specifically.
> This approach could be enriched by taking inspiration from concepts based on methods that perform better on Pneumothorax, such as [Sangjoon et al. (2022)](https://www.nature.com/articles/s41467-022-31514-x) for example.
> Based on this example, we could apply knowledge distillation from a teacher model trained on a Pneumothorax dataset instead of fine-tuning the model directly on a  subset of the data.
> We chose not to apply such methods, since we tried to keep our fine-tuning process as general as possible, to make it easier to compare our method to other approaches. Additionally, we want our approach to be representative for gaining insight into the behavior of a larger family of biomedical vision-language models.
>
> Another option is to work on generating richer reports on Pneumothorax.
> One of the main problems is that, despite the inconsistent nature of this disease, impressions of Pneumothorax rarely contain specific information on size and location.
> More specific medical reports, for example via report generation models, could mitigate this issue and therefore lead to a better localization performance.
>
> **We extended our limitations to address possible future work in regards to challenging diseases such as Pneumothorax.**

---

### Author Response · Authors · 2025-03-07

We would like to thank the reviewers for their insightful feedback and comments. We hope that we could adequately answer open questions and that the changes introduced to our manuscript could address the key problems.
To give an overview, we will shortly summarize feedback we received and based on that, the main changes we introduced to our manuscript.

We are thankful all reviewers agreed that the results of our method show a significant improvement over current state-of-the-art methods in medical phrase grounding, resulting in a relevant contribution to the field from this regard.

However, some concerns were raised regarding the clarity and description of our method. We used this opportunity to considerably enhance and extend the methodological section of our manuscript, to hopefully make the reasoning behind our approach clearer.
Additionally, we provided a further Appendix, that gives additional intuition behind some design choices in our method.

The reviewers were also rightfully concerned regarding the performance of our model on Pneumothorax. As a response, we addressed the issue with this disease more clearly and added potential ways to mitigate the weaker performance regarding this disease.

Per suggestion by one of the reviewers, we extended the discussion of related work, to paint a more coherent picture of the state-of-the-art methods we are comparing our approach to.

To make our results more robust, we evaluated additional training runs of our model, in order to give an estimation for the standard deviation of it.

Thanks to the feedback of one reviewer, we added some details on the computational practicality of our model.

---

### Author Rebuttal · Authors · 2025-03-07

**Rebuttal:**

The supporting material contains our revised manuscript, as well as our revised manuscript with highlighted changes.

**Supporting Material:**

/attachment/974a5b5452fceafaf5b5c3d2966ac1d25aa6d058.zip

---

### Meta-Review · Area_Chair_exgM · 2025-03-22

**Recommendation:** Accept (Poster)
**Confidence:** 3

**Metareview:**

The paper focuses on improving zero-shot grounding performance. Overall, the reviews were thorough, and while they appreciated the idea, had several concerns about the model evaluation metric, results, and computational requirements. Nevertheless, after the author responses, the reviews tilted positive. I think overall there are enough interesting ideas here that they are worth discussing at the conference. I recommend the paper be accepted.